# Effect of an Oral Bivalent Vaccine on Immune Response and Immune Gene Profiling in Vaccinated Red Tilapia (*Oreochromis* spp.) during Infections with *Streptococcus iniae* and *Aeromonas hydrophila*

**DOI:** 10.3390/biology11091268

**Published:** 2022-08-26

**Authors:** Md Shirajum Monir, Md Sabri Mohd Yusoff, Mohd Zamri-Saad, Mohammad Noor Azmai Amal, Aslah Mohamad, Mohamad Azzam-Sayuti, Md Yasin Ina-Salwany

**Affiliations:** 1Department of Aquaculture, Faculty of Agriculture, Universiti Putra Malaysia, Serdang 43400 UPM, Selangor, Malaysia; 2Bangladesh Fisheries Research Institute (BFRI), Mymensingh 2201, Bangladesh; 3Department of Veterinary Pathology and Microbiology, Faculty of Veterinary Medicine, Universiti Putra Malaysia, Serdang 43400 UPM, Selangor, Malaysia; 4Aquatic Animal Health and Therapeutics Laboratory, Institute of Bioscience, Universiti Putra Malaysia, Serdang 43400 UPM, Selangor, Malaysia; 5Department of Biology, Faculty of Science, Universiti Putra Malaysia, Serdang 43400 UPM, Selangor, Malaysia

**Keywords:** oral bivalent vaccine, immune gene expression, post-challenges, red tilapia

## Abstract

**Simple Summary:**

Streptococcosis and aeromonasis are major bacterial diseases in tilapia cultures, causing mass mortality with substantial economic losses to global aquaculture. The development of oral monovalent vaccines to prevent these diseases has been attempted; still, the mechanism of immunity of oral bivalent vaccines against streptococcosis and aeromonasis infections, as well as the protective ability of the host against these two diseases, remains understudied. To explore the immunological role of an oral bivalent vaccine, we compared the immune responses and immune gene profiling in red tilapia post-challenged with *Streptococcus iniae* and *Aeromonas hydrophila*. Our results showed that the degrees of expression of different innate and adaptive immune-associated genes in both mucosal and systemic immune organs were significantly higher in the bivalent vaccinated fish compared to the monovalent and control (unvaccinated) groups.

**Abstract:**

Streptococcosis and aeromonasis inflicted by *Streptococcus iniae* and *Aeromonas hydrophila*, respectively, have affected tilapia industries worldwide. In this study, we investigated antibody responses and explored the mechanisms of protection rendered by an oral bivalent vaccine in red tilapia following challenges with *S. iniae* and *A. hydrophila*. The results of specific IgM antibody response revealed that the IgM titers against *S. iniae* and *A. hydrophila* in the bivalent incorporated (BI) vaccine group were significantly higher (*p* < 0.05) than those in the bivalent spray (BS) vaccine fish and unvaccinated control fish throughout the experiment. Real-time qPCR results also showed that the gene expression of CD4, MHC-I, MHC-II, IgT, C-type lysozyme, IL-1β, TNF-α, and TGF-β remained significantly higher (*p* < 0.05) than that of the controls between 24 and 72 h post-infection (hpi) in both mucosal (hindgut) and systemic (spleen and head–kidney) organs of BI vaccinated fish. Furthermore, the highest relative expression of the TGF-β, C-type lysozyme, and IgT genes in the BI vaccinated group was observed in the challenged fish’s spleen (8.8-fold), head kidney (4.4-fold), and hindgut (19.7-fold) tissues, respectively. The present study suggests that the bivalent incorporated (BI) vaccine could effectively improve the immune function and activate both humoral and cell-mediated immunities in vaccinated red tilapia following the bacterial challenges.

## 1. Introduction

Tilapia (*Oreochromis* sp.) is one of the most important freshwater fish genera that are cultured intensively all over the world. However, with the rapid development of tilapia aquaculture, various infectious bacterial diseases have plagued tilapia production—most significantly, streptococcosis and aeromonasis [1,2]. Streptococcosis is caused by Gram-positive bacteria of *Streptococcus iniae*, while aeromonasis is mainly caused by Gram-negative bacteria of *Aeromonas hydrophila*, and these diseases cause severe economic losses—especially in the tilapia industry [3,4,5]. Hence, the rise in the occurrence of these bacterial diseases creates an urgent requirement for effective preventive measures to resist these presently notorious pathogens.

Fish vaccination is the most effective technique to resist bacterial infections in aquaculture. Some monovalent vaccines have been developed to combat *S. iniae*, *S. agalactiae* [6,7,8,9], and *A. hydrophila* [10,11]. These developed monovalent vaccines impart high protective properties against individual target bacteria through injection or immersion immunization, but they are unable to protect against multiple bacteria through a single vaccination. In contrast, considering the cost and timing of fish vaccination, developing oral bivalent vaccines to prevent these diseases in a single immunization is desirable [12,13]. Moreover, few experimental oral monovalent vaccines are available against streptococcosis infections, with various protections [14,15]. These developed monovalent oral vaccines protect against single bacterial species; however, the exact mechanisms of action are unknown in the oral bivalent vaccines. Therefore, to improve the efficacy of an oral bivalent vaccine, it is crucial to comprehend the effectiveness of immune protection in inoculated fish following a challenge with certain bacteria.

In addition to the vaccine, adjuvants play a significant role in eliciting more robust immune responses. Various mineral oils are broadly applied as adjuvants in fish vaccine formulation, e.g., Freund’s complete or incomplete adjuvants and Montanide adjuvant which are very effective in fish vaccination, with low toxicity levels [16,17,18], but can be costly—particularly for the industrial fish vaccine formulation [19]. Palm oil, an adjuvant used in an oral vaccine against *S. agalactiae,* has been observed to stimulate immune responses in tilapia [20]. Nevertheless, palm oil might be another possible adjuvant in fish vaccine formulation, which may induce effective immunity and provide high protection at a lower cost than the commercial adjuvants.

Expression analysis of immune-related genes in oral-vaccinated fish after challenges has always been of interest in the fields of infection and immunity. An effective vaccination simultaneously activates innate immune responses and specific humoral and cell-mediated immunities, while the cell-mediated immunities are presumed to be predominantly responsible for protecting the fish against infections [13]. However, among the different cytokines, pro-inflammatory cytokines were observed to be elicited in the different organs of vaccinated fish during the initial period of infections with *A. hydrophila* and *S. agalactiae* [6,21]. Stimulation of antigen-presenting molecules, viz., MHC-I and MHC-II, along with the CD4 and CD8 genes—particularly in fish vaccinated through injection and post-challenge fish—have been documented [13,21]. Moreover, the role of specific immunoglobulin of IgT is well documented in post-vaccination rainbow trout [22], but the IgT gene has rarely been studied compared to IgM and IgD [23]—mainly in vaccinated tilapia after bacterial infection.

Previously, we developed and tested an oral bivalent vaccine that has proven effective in protecting red tilapia—*Oreochromis niloticus* × *Oreochromis mossambicus*—from streptococcosis and aeromonasis [24]. The oral bivalent vaccine was found to give high protection to the host, with a relative percent survival (RPS) of 82.22 ± 3.85% post-*S. iniae* challenge, 77.78 ± 3.85% post-*A. hydrophila,* challenge, and 77.78 ± 3.85% after being challenged with the mixture of *S. iniae* and *A. hydrophila*. The bivalent vaccination treatment also showed significantly greater partial cross-protections (*p* < 0.05) following *S. agalactiae* (RPS at 60.00 ± 6.67%) and *A. veronii* (RPS at 57.78 ± 7.70%) challenges. However, the information about the induction of immune gene expression in mucosal and systemic immune tissues after experimental infections in oral bivalent vaccinated fish is minimal. Therefore, the present study aimed to describe the antibody (IgM) response and analyze immune gene expression to understand the immune genes’ interactions in various immune organs, along with the immune responses involved in the protection of oral bivalent vaccinated red tilapia against streptococcosis and aeromonasis.

## 2. Materials and Methods

### 2.1. Bacterial Strain and Growth Condition

*Streptococcus iniae* and *Aeromonas hydrophila* strains were isolated from moribund red tilapia (*Oreochromis* spp.) during streptococcosis [3] and aeromonasis [19] outbreaks at different tilapia farms in Malaysia. These two isolates were independently cultured on tryptone soy agar (TSA, Oxoid, Basingstoke, UK) on plates for 28 h at 32 °C. Afterwards, the *S. iniae* and *A. hydrophila* isolates were grown individually in 200 mL of TSB broth and cultured for 28 h with gentle shaking at 32 °C.

### 2.2. Ethical Approval

All procedures involving animals in this study were compliant with the guidelines of the Malaysian Code of Practice for The Care and Use of Animals for Scientific Purposes, and approved by the Institutional Animal Care and Use Committee, Universiti Putra Malaysia (approval number #UPM/IACUC/AUP-R076/2019).

### 2.3. Fish Maintenance

Healthy red tilapia weighing 31.27–34.25 g were purchased from a commercial fish hatchery (Selangor, Malaysia) and transferred to the Hatchery Unit, Institute of Bioscience (IBS), Universiti Putra Malaysia (UPM). The fish were reared under laboratory conditions for 14 days prior to the start of the experiment. The red tilapia were fed commercial feed (Star Feed, Malaysia) at 3% of body weight daily. The water quality was checked through the YSI multiprobe system 556: temperatures at 27–29 °C, dissolved oxygen at 5.5–6.5 mg/L, pH at 6.8–7.7, and ammonia nitrogen < 0.01 mg/L.

### 2.4. Feed-Based Vaccine Preparation

*Streptococcus iniae* and *A. hydrophila* bacteria were grown independently in TSB broth and incubated at 130 rpm for 28 h at 32 °C. The formalin-killed cells (FKCs) (bacterins) were prepared based on previous studies [14,15]. Briefly, the bacterial suspensions were treated with 0.5% formalin and kept for 24 h at 4 °C. The killed bacterial suspensions were centrifuged at 6000× *g* for 15 min. The bacterial pellets were washed three times with sterile PBS, and then the concentration of FKCs was adjusted to ~10^9^ CFU/mL and kept at −20 °C until used. The bivalent vaccines were formulated with a mixture of killed cells of *S. iniae* and *A. hydrophila* vaccines at a ratio of 1:1 (*v*/*v*), and maintained at −20 °C until further use. Subsequently, an adjuvant (10% palm oil) was added [20] before it was sprayed on the feed to adjust the concentration to ~10^9^ cells/g of feed [25].

The vaccines were prepared following the methods described by Ismail et al. [15], with some modifications. For the feed-based bivalent spray (BS) vaccine, the prepared bivalent vaccine solution with 10% palm oil was top-dressed on fish feed pellets to achieve an individual final FKC concentration of ~10^9^ cells/g of feed (*S. iniae* containing ~10^9^ cells and *A. hydrophila* containing ~10^9^ cells/g of feed). Moreover, the bivalent vaccine and monovalent *S. iniae* or *A. hydrophila* vaccine with 10% palm oil were incorporated independently into the fish feed powder to maintain a final concentration of ~10^9^ cells/g of feed to prepare bivalent incorporated (BI) or monovalent vaccines. Then, the vaccine-mixed feed was re-pelleted using feed pellets and followed by dry heating at 28 °C for 12 h using a convection oven. Finally, the prepared feed-based vaccines were stored at 4 °C until use. In order to assess the safety of the prepared vaccines, healthy fish were fed the vaccinated feed pellets at 5% of body weight per day for five consecutive days. No disease symptoms were noted in any of the feed-based vaccinated fish within 14 days.

### 2.5. Feed-Based Vaccination

Following a two-week acclimation period, 1470 hybrid red tilapia were randomly dispersed into five groups, and each group consisted of three replicates (98 fish/replicate). BS group: fish were immunized using the bivalent spray feed vaccine (i.e., whole-cell inactivated *S. iniae* and *A. hydrophila* coated on feed); BI group: fish were immunized using the bivalent incorporated feed vaccine (i.e., whole-cell inactivated *S. iniae* and *A. hydrophila* combined into feed); MS group: fish were immunized using the monovalent *S. iniae* feed vaccine; MA group: fish were immunized using the monovalent *A. hydrophila* feed vaccine; and control group: commercial feed with 10% palm oil. Fish were not given any feed a day before the immunization. Prior to the vaccinations, the feed-based vaccines were orally administered to all immunization treatments on day 0 at 5% of body weight daily for five consecutive days. Booster doses were given on days 14 and 42 following the first vaccination (Figure 1). On other days, all tilapia were fed with commercial pellets for the duration of the 112-day experiment.

### 2.6. Challenge Tests

On the 70 days post-vaccination, the BS, BI, and MS immunized groups and the control (unvaccinated) group were intraperitoneally (i.p.) injected with 0.5 mL of *S. iniae* (3.4 × 10^9^ CFU/mL). Meanwhile, the BS, BI, MA, and control groups were i.p. injected with 0.5 mL of *A. hydrophila* (6.8 × 10^9^ CFU/mL). Fish were anesthetized by immersion in water containing 100–106 mg/mL of tricaine methanesulfonate (MS-222, Sigma-Aldrich, St. Louis, MO, USA) before collecting the fish samples. In the challenge tests, the immunized and control challenged fish were triplicated, and each replicate had 15 fish. Mortality patterns and clinical signs of the infected fish were checked daily for 14 days post-infection, and dead fish were examined for *Streptococcus*- or *Aeromonas*-specific mortalities.

### 2.7. Sampling

Gut lavage fluid samples from 6 fish per group were collected biweekly from the experimental fish. The gut lavage fluid was stored in tubes containing 100 μL of PBS with 0.02% (*w*/*v*) sodium azide. The collected lavage fluid was then refrigerated and centrifuged at 3000× *g* for 3 min, and the supernatant was kept at −20 °C until further assessment of the specific antibody (IgM) levels.

The spleen, head kidney, and hindgut tissues from six fish per group were collected at 12, 24, 48, 72, and 96 h post-infection (hpi), and preserved in RNAlater (Invitrogen, Waltham, MA, USA) at −80 °C until the RNA was isolated. Moreover, the brains, kidneys, spleens, and livers from 30 challenged fish (6 fish/group) were preserved in 10% buffered formalin for histopathological analysis.

### 2.8. Specific Antibody Detection by Enzyme-Linked Immunosorbent Assay (ELISA)

Antibody (IgM) titers in red tilapia gut lavage fluid against *S. iniae* and *A. hydrophila* were evaluated following the standard indirect ELISA technique [14]. Briefly, a flat-bottomed microplate (Sigma, St. Louis, MO, USA) was coated independently with 2.7 × 10^5^ CFU/mL of *S. iniae* or *A. hydrophila* in 100 μL/well coating buffer (pH 9.6) and maintained at 4 °C overnight. The bacteria-coated microplates were washed in PBS with 0.05% Tween-20 (PBST), followed by 200 μL of 1% bovine serum albumin (PBS + 0.05% Tween-20 + BSA, Sigma, USA) at 37 °C for 1 h. After blocking, gut lavage fluid from each vaccinated and control fish was diluted separately (1:1000) in PBST and added to the wells (100 μL per well) in triplicate. The wells were washed thrice with PBST after incubation, followed by adding 100 μL of goat anti-tilapia hyperimmune serum (1:10,000 dilution in PBST) to each well, and were kept at 37 °C for 1 h. Following the incubation, 100 μL/well rabbit anti-goat IgM-horseradish peroxidase (diluted to 1:10,000 dilution in PBST, Nordic Immunological Laboratory, Eindhoven, The Netherlands) was added after washing three times with PBST. Then, 100 μL/well TMB (Promega, Madison, MI, USA) was added and incubated at 37 °C for 30 min; 50 μL of stop solution (2.5 M H_2_SO_4_) was added to each well of the plates, and the absorbance was measured at OD 450 nm with a microtiter plate reader (Thermo Fisher, Waltham, MA, USA).

### 2.9. RNA Extraction, cDNA Synthesis, and RT-qPCR

Following the challenge tests, experimental fish spleen, head kidney, and hindgut tissues were sampled for total RNA extraction using TRIzol Reagent (Invitrogen™, Thermo Fisher Scientific, Carlsbad, CA, USA) as per the manufacturer’s instructions. The RNA purity was checked based on absorbance at an OD 260 nm/OD 280 ratio using the Genova Nano (Jenway, Staffordshire, UK). Using a reverse transcription kit (QIAGEN, Hilden, Germany), one microgram of total RNA was used to create the single-stranded cDNA. RT-qPCR was conducted as described in the manufacturer’s protocol of SYBR Green RT-qPCR mixture (QIAGEN, Hilden, Germany) using Rotor-Gene RT-qPCR (QIAGEN, Hilden, Germany), and each sample consisted of three technical replicates for RT-qPCR evaluation. The primers for each gene used for qPCR assays are presented in Table 1. All qPCR assays were standardized at an annealing temperature of 59 °C, and had efficiencies within 100% ± 4%. The relative expressions of the immune-related genes were calculated through the threshold cycle method (2^−ΔΔCt^ method), with β-actin as the housekeeping gene [6].

### 2.10. Statistical Analysis

IBM SPSS Statistics 26 (SPSS 26.0 package, SPSS Inc., Chicago, IL, USA) was used to perform the statistical analyses. One-way ANOVA and Duncan’s new multiple range test were used to test the significance of differences observed between the vaccinated and control groups. The significance levels were set as *p* < 0.05. The results are shown as the mean ± SE.

## 3. Results

### 3.1. IgM Titers in Gut Lavage Fluid

In gut lavage fluid, the anti-*S. iniae* IgM titers of all vaccinated groups gradually increased and were significantly higher (*p* < 0.05) than those of the control group from 14 to 70 days post-vaccination (Figure 2A). The specific IgM antibody in the control treatment was not detected, with almost 100% mortality following the challenge test. In addition, significantly (*p* < 0.05) greater IgM titers against *S. iniae* were shown in the BI and MS groups compared with the BS group from 84 to 112 days post-vaccination. Moreover, the anti-*A. hydrophila* IgM titer in the gut lavage fluid of all vaccinated fish was also significantly higher (*p* < 0.05) than that of the unvaccinated fish from 14 to 70 days post-vaccination (Figure 2B). At the same time, the IgM titer against *A. hydrophila* induced in the BI and MA groups was significantly higher (*p* < 0.05) than that in the BS group at 84- and 98-days post-vaccination, but the difference in titers between the BI and MS groups was not significant except for 28 and 56 days post-vaccination.

### 3.2. Expression Profiles of Immune-Related Genes Following Challenge with Streptococcus iniae

In the spleen after challenge with *S. iniae*, the relative gene expression of IL-1β in the BI and MS groups showed a significantly higher peak (*p* < 0.05) at 24 h post-infection (hpi) (8.2-fold in BI- and 5.1-fold in MS-vaccinated fish) than the unvaccinated treatment (Figure 3A). The expression of the C-type lysozyme gene in fish vaccinated with the BI vaccine had a significantly (*p* < 0.05) higher peak at 24 hpi (7.1-fold), while the BI and MS groups’ expression levels insignificantly (*p* > 0.05) decreased at 96 hpi (Figure 3B). The expression levels of TNF-α in the BI- and MS-vaccinated fish were significantly higher (*p* < 0.05) at 24 and 48 hpi, but the BI group had significantly higher expression at 24 hpi (8.9-fold) than the other groups (*p* < 0.05) (Figure 3C). The TGF-β gene expression in the BI and MS groups was significantly higher (*p* < 0.05) at 96 hpi, but the MS group statistically (*p* < 0.05) showed the highest expression when compared to the other groups (Figure 3D). In BI- and MS-vaccinated fish, the expressions levels of the MHC-I and CD4 genes increased progressively at 12 hpi, and reached a significant (*p* < 0.05) peak only in the BI group at 24 hpi (9.1-fold) for CD4, but at 48 hpi (11.2-fold) for the MHC-I gene (Figure 4A,B). By 24 and 48 hpi, MHC-II expression in all vaccinated groups was found to be significantly greater (*p* < 0.05), and the highest level of expression in the BI group was noted at 24 hpi (Figure 4C). In the case of the IgT gene, all of the vaccinated groups had significantly greater expression (*p* < 0.05) at 72 and 96 hpi, but the highest peak (*p* < 0.05) was in the BI vaccine treatment at 72 hpi (14.1-fold) (Figure 4D).

In the head kidney, the highest expression of both the IL-1β and C-type lysozyme genes at 24 hpi (5.1-fold for IL-1β and 4.4-fold for C-type lysozyme) was shown in the incorporated bivalent vaccine treatment compared to the other groups (Figure 5A,B). The vaccinated groups showed slight expression of the TNF-α gene at 12 hpi, but significant (*p* < 0.05) induction at 48 hpi, followed by a peaked induction at 48 hpi (5.7-fold) (Figure 5C). Both the BI- and MS-vaccinated groups demonstrated significantly lower TGF-β gene expression at 48 hpi (*p* < 0.05), but significantly (*p* < 0.05) higher expression was found at 96 hpi (2.4-fold) compared to the other groups (Figure 5D). The CD4 expression in the BI group reached a peak at 72 hpi (5.6-fold) compared to the other groups while surprisingly, the MS group had significantly (*p* < 0.05) lower expression at 96 hpi (Figure 6A). The MHC-I gene expression was significantly higher (*p* < 0.05) in the immunized groups at 48 hpi, among which the BI group showed significantly (*p* < 0.05) the greatest expression at the same time point (10.38-fold) (Figure 6B). The vaccinated groups showed slight expression of the MHC-II gene at 12 hpi, but significantly (*p* < 0.05) elevated expression at 24 hpi, followed by a peaked induction in the BI group at 48 hpi (9.9-fold) (Figure 6C). Furthermore, the IgT gene was significantly elevated (*p* < 0.05) in the BI- and MS-vaccinated red tilapia at 72 and 96 hpi; subsequently, the BI group showed the highest expression (*p* < 0.05) of IgT at 96 hpi (6.3-fold) compared to the other groups (Figure 6D).

In the hindgut, the IL-1β gene expression in the BI group was significantly higher (*p* < 0.05) at 24 hpi (10.9-fold) (Figure 7A). The expression levels of the C-type lysozyme gene in the BI and MS groups were significantly higher from 12 to 48 hpi (*p* < 0.05), while the BI group showed significantly (*p* < 0.05) higher expressions at 24 hpi (8.5-fold) compared to the other groups (Figure 7B). The relative expression of TNF-α in the bivalent incorporated vaccine group peaked at 24 hpi (10.8-fold) but, surprisingly, only the MS-vaccinated group showed significantly (*p* < 0.05) lower expression than the other groups at 96 hpi (Figure 7C). In the immunized treatments, significantly lower TGF-β expression was observed at 48 and 72 hpi (*p* < 0.05), but insignificantly (*p* > 0.05) higher expression was noticed at 96 hpi compared to the control group (Figure 7D). The expression levels of the CD4 gene in the vaccinated treatments were significantly (*p* < 0.05) higher at 24 and 48 hpi, whereas the MS group was found to be significantly (*p* < 0.05) higher and peaked at 48 hpi (8.1-fold) (Figure 8A). In the case of the MHC-I gene, only the BI group had significantly higher (*p* < 0.05) expression at 48 hpi (11.7-fold) (Figure 8B). Surprisingly, the MHC-II gene at 12 hpi in the MS group demonstrated a lower expression compared to the other groups, but the BI and MS treatments demonstrated a significantly greater expression (*p* < 0.05) at 48 and 72 hpi compared to the unvaccinated treatment. In addition, the MHC-II gene expression in the BI group was significantly (*p* < 0.05) higher at 72 hpi (12.3-fold) compared to the control group (Figure 8C). Furthermore, the IgT gene expression in the BI vaccine treatment was the highest (*p* < 0.05) at 72 hpi (19.7-fold) as compared with the other groups (Figure 8D).

### 3.3. Expression Profiles of Immune-Related Genes Following Challenge with Aeromonas hydrophila

In the spleen, after being infected with *Aeromonas hydrophila*, the expression of the IL-1β gene in the BI- and MA-vaccinated fish was found to be significantly higher (*p* < 0.05) between 12 and 72 hpi, while the maximum expression in the BI group was observable within 24 hpi (2.73-fold) (Figure 9A). The treated fish were found to have a significantly higher (*p* < 0.05) expression of the C-type lysozyme gene from 12 to 48 hpi, whereas the MA treatment was significantly (*p* < 0.05) lower at 96 hpi than that of the other groups (Figure 9B). Both the BI and MA treatments had significantly (*p* < 0.05) higher expression of the TNF-α gene at 24 hpi (6.9-fold in the BI group and 10.4-fold in the MA group) compared to the control group (Figure 9C). The BI and MS immunized groups had significantly (*p* < 0.05) lower expression of the TGF-β gene at 24 hpi, but higher expression was noted in both of the vaccinated groups at 72 hpi compared to the unvaccinated group (Figure 9D). Between 24 and 72 hpi, the CD4 and MHC-I gene expression in all immunized treatments was greater (*p* < 0.05) than in the unvaccinated treatment (Figure 10A,B), but the bivalent incorporated vaccine group was the only one to show the highest expression of CD4 (8.6-fold) at 24 hpi and MHC-I (11.9-fold) at 48 hpi (*p* < 0.05). The MHC-II gene expression in the BI- and MA-vaccinated groups was significantly high (*p* < 0.05), and peaked at 24 hpi, as compared to the unvaccinated group (Figure 10C). In the case of the IgT gene, the maximum expression in the BI and MA immunized treatments was observed at 72 hpi (12.1-fold in the BI group and 11.5-fold in the MA group), which was significantly higher than that of the unvaccinated group (Figure 10D).

In the head kidney, the gene expression of IL-1β and C-type lysozyme in the BI vaccine group was significantly higher (*p* < 0.05) at 24 hpi than in the other groups (Figure 11A,B). The gene expression levels of TNF-α in BI- and MA-vaccinated fish peaked at 24 hpi (6.1-fold in the BI group and 5.9-fold in the MA group) while, surprisingly, both the BI and MS groups had significantly lower expression (*p* < 0.05) at 96 hpi (Figure 11C). In the case of the TGF-β gene, lower expression in the BI- and MA-vaccinated groups was noticed at 24 hpi (*p* < 0.05), but higher expression was found in the BI group at 72 hpi (2.6-fold) compared to the unvaccinated group (*p* < 0.05) (Figure 11D). The expression of the CD4 and MHC-I genes was significantly high (*p* < 0.05) only in the immunized red tilapia from the BI group at 48 hpi (3.7-fold) for CD4, but at 24 hpi (7.2-fold) for the MHC-I gene when compared with the other groups (Figure 12A,B). Both BI- and MA-vaccinated fish showed significantly higher expression of the MHC-II gene at 24 hpi than the control group (*p* < 0.05) while, surprisingly, expression of this gene was significantly decreased (*p* < 0.05) in the MA immunization treatment at 96 hpi (Figure 12C). The expression of the IgT gene in the BI group peaked at 72 hpi (9.2-fold), and was significantly higher than in the other groups (*p* < 0.05) (Figure 12D).

In the hindgut, the highest IL-1β and C-type lysozyme gene expression levels were found at 24 hpi in the BI vaccine group, which was significantly higher than in the other treatments (*p* < 0.05) (Figure 13A,B). The expression levels of the TNF-α gene were significantly (*p* < 0.05) increased in the BI and MA immunization treatments at 12 to 48 hpi, and reached a peak at 24 hpi (10.1-fold in the BI group and 8.7-fold in the MA group) relative to the control group (Figure 13C). Moreover, the expression levels of the TGF-β gene were found to be significantly (*p* < 0.05) lower at 48 hpi in both the BI and MS groups, while only the fish from the bivalent incorporated vaccine treatment group reached significantly higher (*p* < 0.05) TGF-β expression at 72 hpi (2.19-fold) compared to the other groups (Figure 13D). The expression levels of the CD4 and MHC-I genes in the BI- and MS-vaccinated treatments were higher than those in the other treatments (*p* < 0.05) at 24 to 72 hpi, and the BI vaccine group reached a peak at 48 hpi (Figure 14A,B). Furthermore, both the BI- and MA-vaccinated groups had significantly (*p* < 0.05) higher expression of the MHC-II gene at 48 hpi (8.1-fold in the BI group and 7.9-fold in the MA group) than the unvaccinated group (Figure 14C). Surprisingly, the BI- and MA-vaccinated groups showed insignificantly (*p* > 0.05) lower expression of the IgT gene at 12 hpi compared to the control group, but the BI vaccine treatment had the highest expression of the IgT gene at 72 hpi (17.4-fold) compared to the other groups (Figure 14D).

### 3.4. Protection of the Feed-Based Vaccinated Red Tilapia

Exophthalmia, ocular opacity, bleeding in the brain, enlarged and hemorrhagic spleen and kidney, irregular swimming (spinning), and exophthalmia were the predominant clinical indications of moribund red hybrid tilapia after challenge with *S. iniae*, while *A. hydrophila*-challenged moribund fish showed hemorrhage with lesions on the body, scale protrusion, and severe hemorrhage in liver and kidney tissues. Interestingly, *S. iniae* and *A. hydrophila* were recovered independently from the dead fish after being challenged with *S. iniae* and *A. hydrophila,* respectively, indicating that mortality resulted from the challenge trials. Nevertheless, the BI-vaccinated group exhibited a more significant relative percentage of survival (RPS) (82% and 77% post-infection with *S. iniae* and *A. hydrophila,* respectively) than the other vaccinated treatments (*p* < 0.05).

### 3.5. Histopathological Analysis after Experimental Infections

Consistent histological lesions were obtained in unvaccinated red hybrid tilapia brains, kidneys, spleens, and livers following the challenge with *S. iniae* and *A. hydrophila*. The infected control fish brains with *S. iniae* presented thickening of the meninges due to the infiltration of neutrophils and severely congested blood vessels surrounded by mononuclear inflammatory cells and empty space (Figure 15A). The unvaccinated fish kidneys after infection with *S. iniae* showed loss of renal tubule architecture, epithelial tubular necrosis with lysis of the cytoplasm, mononuclear cell infiltration, and necrotic areas between the renal tubules (Figure 15B). The spleens of unvaccinated fish following challenge with *A. hydrophila* showed extensive areas of necrosis and depletion of lymphocytes, vascular congestions surrounded by the inflammatory cell infiltrations, and marked production of melanomacrophage centers (MMC) (Figure 15C). The liver tissues of unvaccinated fish exhibited congestion with focal necrosis of hepatocytes, diffuse mononuclear hepatic inflammation, infiltrations of inflammatory cells at the central vein, and degenerative changes with hepato-cytoplasmic vacuolization after being experimentally infected with *A. hydrophila* (Figure 15D).

## 4. Discussion

Specific antibody production is a significant parameter of the immune response following vaccination and infection, broadly used as a correlate of protection for fish vaccines [9]. The findings of the present study showed that the trend of antibody (IgM) titers against *S. iniae* and *A. hydrophila* was significantly greater in BI-immunized fish (*p* < 0.05), and was maintained until 70 days post-vaccination, suggesting good immunogenicity of the BI-vaccinated group. Following challenges, the BI-immunized tilapia produced higher antibody (IgM) titers against *S. iniae* and *A. hydrophila* from 85 to 112 dpi, compared to the other groups. These findings demonstrate that BI vaccination not only functioned against *S. iniae* bacteria, but also protected against *A. hydrophila* infection. On the other hand, fish vaccine research data suggest broadly accepted views that antibody (IgM) titers are potentially the most reliable correlate of protective immunities induced by fish vaccines such as the *S. iniae* and *S. agalactiae* vaccine [9,30] or the *A. hydrophila* vaccine [31]. In the present study, the RPS of the BS, MS, and MA groups was significantly (*p* < 0.05) higher than that of the BI-vaccinated fish, with 82% and 77% RPS after *S. iniae* and *A. hydrophila* challenges, respectively. These results suggest that both mucosal and systemic antibody (IgM) titers were related to the immune protection against the attacks by these two pathogenic bacteria [13,32], and could ultimately reduce mortality from bacterial challenges [28].

The effectiveness of a vaccine-induced immune response always relies on interaction between the innate and adaptive immune systems, which could prevent the infection by the pathogens [28]. Therefore, many studies have begun to look at the effects of immunization, with a primary focus on events occurring immediately post-immunization after injection or bath vaccination [9,22]. However, no study has focused on early post-infection events in oral bivalent vaccinated fish. Hence, to understand the immunological basis of the oral bivalent vaccines’ efficacy, the present study encompassed the relative expression kinetics of MHC-I, CD4, MHC-II, IgT, C-type lysozyme, IL-1β, TNF-α, and TGF-β immune-related genes in the systemic (i.e., head kidney and spleen) and mucosal (i.e., hindgut) tissues of the vaccinated fish during the post-challenge period.

IL-1β and TNF-α, the classic pro-inflammatory cytokine genes, are induced by activated immune cells upon stimulation by different bacterial, parasitic, and viral antigens, and control the balance between the humoral and cellular immune response [23]. They are induced by T-lymphocytes and mononuclear phagocytes involved in both the adaptive and innate immune pathways [9]. In the present research, higher expression of TNF-α and IL-1β in both systemic and mucosal tissues was induced in the BI-vaccinated groups against *S. iniae* and *A. hydrophila* infection, respectively, illustrating that BI-immunized fish were more effective in activating macrophage and phagocyte activity against bacterial infection compared to the control group [1]. Similar phenomena have also been documented in vaccinated rohu (*L. rohita*) challenged with *A. hydrophila* [33], Nile tilapia (*Oreochromis niloticus*) challenged with *S. agalactiae* [7], shabout fish infected with *A. hydrophila* [34], and grass carp infected with *Flavobacterium columnare* [35]. Additionally, the upregulated levels of the IL-1β and TNF-α genes in the BI vaccine group in this study, up to 48 h post-infection (hpi), suggest that activation of the IL-1β and TNF-α genes occurs immediately after infection, and the decline in their levels over time may reflect a reduction in the bacterial burden or neutralization of the inflammation [21].

As the first line of defense against pathogen invasion, the innate immune system is known to perform a greater protective function in fish than the adaptive immune system [9]. In this study, significantly high expression of C-type lysozyme was noticed in the systemic and mucosal tissues of BI-vaccinated hosts after infection with *S. iniae and A. hydrophila,* respectively. This study’s results are consistent with studies conducted by Jinendiran et al. [36] and Ling et al. [37]. In addition, Pridgeon et al. [38] and He et al. [39] also recorded significantly (*p* < 0.05) greater expression of C-type lysozyme in the guts of vaccinated catfish and the kidneys, spleens, and guts of vaccinated European eels (*Anguilla anguilla*). However, the C-type lysozyme in all of the examined tissues in this study was significantly (*p* < 0.05) expressed post-infection, indicating that the innate system of BI-vaccinated fish was elicited and provided a certain degree of disease resistance and protection against bacterial infections.

The anti-inflammatory cytokine TGF-β is expressed by regulatory T cells, which are related to self-tolerance, tissue homeostasis, autoimmunity, and suppression of immune responses during infection [40]. In the present study, the upregulation of TGF-β was most notable in the head kidney, spleen, and hindgut—particularly in the fish from the BI vaccine group after infection with *S. iniae*. These findings are consistent with the previous studies on TGF-β expression in systemic and mucosal organs infected by bacteria and parasites [40,41]. Furthermore, in this study, the TGF-β increased in post-challenge fish, suggesting that BI-vaccinated fish maintain the dynamic balance between the inflammatory reactions and avoid triggering needless damage to the host [28].

The functional relevance of B and T lymphocytes in the systemic and mucosal immune compartments is central to the fish’s adaptive immune response, which depends on the presentation of antigens by MHC markers available on antigen-presenting cells (APCs) [42]. Therefore, these marker molecules’ expression can help us to assess the post-infection efficacy of oral bivalent vaccines. In the present study, the detected genes involved in the cellular (MHC-I) and humoral immunity (CD4 and MHC-II) in BI-vaccinated fish were upregulated from 48 to 72 hpi compared with the control group. Following infection with *S. iniae* and *A. hydrophila,* the strong upregulation of MHC-I and MHC-II, respectively, in the spleen, head kidney, and hindgut tissues of bivalent incorporated vaccinated fish indicated that the BI vaccine was successfully administered and presented by the MHC receptor. This upregulation could eventually initiate the proliferation of MHC-II and CD4 B cells, along with MHC-I- and CD8-T-cell-mediated cytotoxic immune responses in the vaccinated fish’s systemic and mucosal tissues [21]. In previous studies, significant upregulation of CD4 and MHC-II has also been observed in the spleen and gut of vaccinated tilapia [28]; the gills, skin, and liver of vaccinated Asian seabass [43]; and the skin, intestines, spleen, and kidneys of vaccinated flounder fish [44], suggesting that both mucosal and systemic immune cells are directly connected in humoral immune responses. However, our study found that the BI vaccine could significantly induce red tilapia’s cellular and humoral immune responses, and evoked high post-infection protection with *S. iniae* and *A. hydrophila*.

The main humoral components of the fish’s adaptive immune system are immunoglobulins (Igs), which are expressed as B-lymphocyte receptors or produced in plasma and mucus [42]. Among the two Ig isotypes identified in fish, IgM is more prevalent in the systemic system, whereas IgT is more prominent in the mucosal compartment [42]. In the present study, the IgT expression in the head kidney, spleen, and hindgut tissues of the bivalent incorporated immunized treatment was significantly upregulated in a time-dependent manner, and reached the peak in hindgut tissues between 72 and 96 hpi, indicating that IgT might contribute to the immediate protection of the host against *S. iniae* or *A. hydrophila* infection at the mucosal surface. It is worth noting that the relatively high IgT expression (19.65-fold) in the hindgut at 72 hpi was primarily due to shallow basal expression of IgT in the unvaccinated fish. The significant upregulation of IgT at early time points is consistent with the hypothesis that IgT generating B lymphocytes is present locally, resulting in stimulated expression of this immunoglobulin upon induction by vaccinated fish after infection [28]. In agreement with these results, a significant level of IgT was expressed in the spleen, liver, and head kidney tissues of vaccinated rainbow trout after being challenged with *Yersinia ruckeri* [40]. Another study by Pérez-Sánchez et al. [45] showed that IgT was also dominant, and found upregulation of IgT in the gut and the head kidney of immunized rainbow trout after challenge with *Lactococcus garvieae*, which is consistent with our findings. Nevertheless, the upregulation of the IgT gene in the head kidney, spleen, and hindgut of immunized fish suggests that IgT is functional in mucosal and systemic immune responses in oral BI-vaccinated fish.

The results of this study demonstrate that BI-vaccinated fish showed greater upregulation of cytokines (TNF-α, TGF-β, and IL-1β), immune cell receptors (CD4, MHC-I, and MHC-II), and immunoglobulin (IgT) genes compared to other vaccinated and control groups. This supports a corresponding positive correlation between upregulation of immune-related genes and specific antibody (IgM) levels in BI-vaccinated fish in this study. Meanwhile, unvaccinated fish in this study had deficient immune gene expression and antibody levels lower than the vaccinated fish, proving that they offered no protection to the infected fish. It is therefore also clear that the immune gene expression and antibody (IgM) levels of all orally vaccinated groups in the present study were directly proportional to the relative percent survival (RPS) values.

Additionally, a study by Yao et al. [28] indicated a high correlation between adaptive gene expression and antibody levels in tilapia after vaccination with inactivated *S. agalactiae* via the oral route. As studied by Jaafar et al. [40] and Xu et al. [23], they too linked the increase in immune gene expression and antibody levels with the increase in RPS values, and observed a strong correlation between the immune-related gene expression, specific antibody (IgM) levels, and RPS in the vaccinated fish. Nevertheless, we found that the oral BI vaccine in this study could promote significantly (*p* < 0.05) higher expression of genes involved in both humoral and cellular immunity, and enhanced the specific antibodies in the vaccinated fish as compared with the other vaccinated and unvaccinated groups.

The bivalent incorporated (BI)-vaccinated fish in this study had the highest IgM antibody levels and upregulation of cytokines, immune cell receptors, and immunoglobulin (IgT) genes, closely followed by the MS and MA vaccine groups. However, the oral bivalent BI vaccine showed higher antibody (IgM) levels and survival rates than the monovalent MS and MA vaccines. This might be because the bivalent antigen’s synergistic effect means higher immunogenicity than the two individual antigens [46]. These results support previous findings that demonstrated the possible superiority of bivalent antigen vaccines as opposed to single-antigen vaccines to achieve more immunocompetent effects [46]. Moreover, the antibody (IgM) levels—particularly in BI-vaccinated fish—remained above those of the unvaccinated group until the end of the experiment on day 112. Thus, the BI vaccine in the present study was able to prolong the immunity by a minimum of four months, and was suggested to contribute significantly to the greatest protection against pathogenic *S. iniae* and *A. hydrophila* infections.

The fish infected with *S. iniae*—particularly those from the unvaccinated (control) group—showed many histopathological changes in the brain caused by streptococcosis, including meningoencephalitis with hemorrhages, edema, and hypertrophy nuclei, all of which are distinctive indicators of infection by streptococcosis [47]. The histopathological examination of the infected internal organs of unvaccinated fish (e.g., kidneys, spleen, and liver) also showed significant pathological changes, such as severe hemorrhage, congestion, infiltration of eosinophilic granular inflammatory cells, necrosis, vacuolation, hemosiderin deposition, melanomacrophage aggregations, and ruptured veins after infections. Ortega et al. [48] (2018) described the histopathology of tilapia, and documented multifocal interstitial nephritis in the kidney, capillary congestion, hemosiderin deposits, melanomacrophage centers in the spleen, and multifocal granulomatous necrosis in the infected liver with *Streptococcus* sp. and *Aeromonas* sp. However, these results are also consistent with the progression of bacterial septicemia, with similar histological descriptions existing for *Streptococcus* sp. or *Aeromonas* sp. bacteria in tilapia [48].

## 5. Conclusions

In this study, upregulation of innate and adaptive genes at early stages post-infection, followed by increases in systemic and mucosal antibody (IgM) titers, along with positively correlated protections, could serve as efficient markers in the evaluation of vaccine efficacy. In addition, the oral bivalent incorporated (BI) vaccine stimulated significant humoral and cellular immune responses, and provided strong protection against virulent *S. iniae* and *A. hydrophila* challenges in red tilapia. Nevertheless, more research on this oral bivalent vaccine formulation is necessary before it can be deployed in the field to potentially reduce the mortality inflicted by *S. iniae* and *A. hydrophila* in the aquaculture industry—especially in tilapia cultures.

## Figures and Tables

**Figure 1 biology-11-01268-f001:**
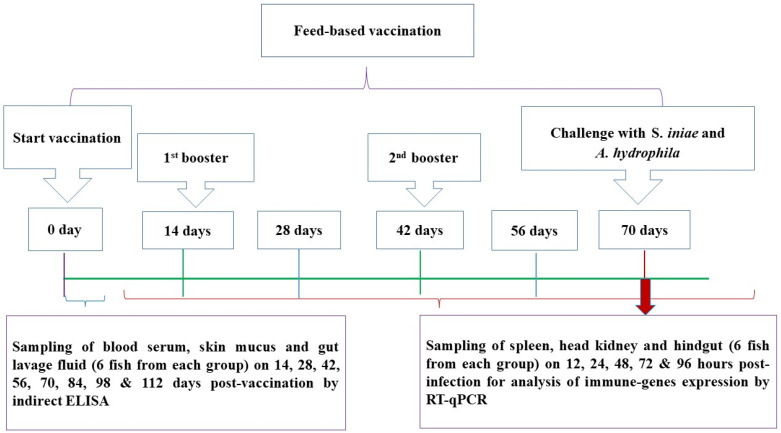
Timeline of the feed-based vaccination regime and challenge assays in red hybrid tilapia.

**Figure 2 biology-11-01268-f002:**
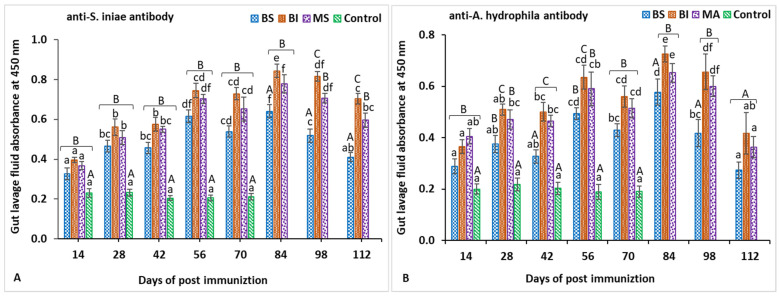
Gut lavage fluid antibody titers against *Streptococcus iniae* (**A**) and *Aeromonas hydrophila* (**B**) on different days post-immunization. Data are presented as the mean ± SE of six red tilapia. Statistically significant differences (*p* < 0.05) on the same days are denoted by capital letters, whereas significant differences between days are denoted by small letters.

**Figure 3 biology-11-01268-f003:**
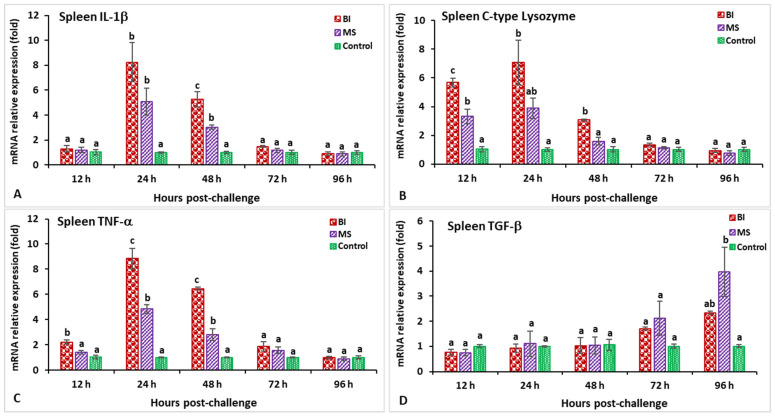
The gene expression of IL-1β (**A**), C-type lysozyme (**B**), TNF-α (**C**), and TGF-β (**D**) in the spleens of orally vaccinated fish after being challenged by *Streptococcus iniae* (mean ± SE, *n* = 3). Dissimilar letters denote significant differences (*p* < 0.05).

**Figure 4 biology-11-01268-f004:**
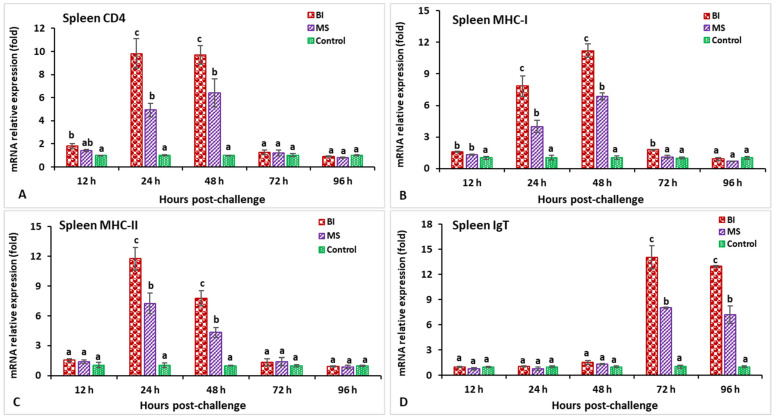
The gene expression of CD4 (**A**), MHC-I (**B**), MHC-II (**C**), and IgT (**D**) in the spleens of orally vaccinated fish after being challenged by *Streptococcus iniae* (mean ± SE, *n* = 3). Dissimilar letters denote significant differences (*p* < 0.05).

**Figure 5 biology-11-01268-f005:**
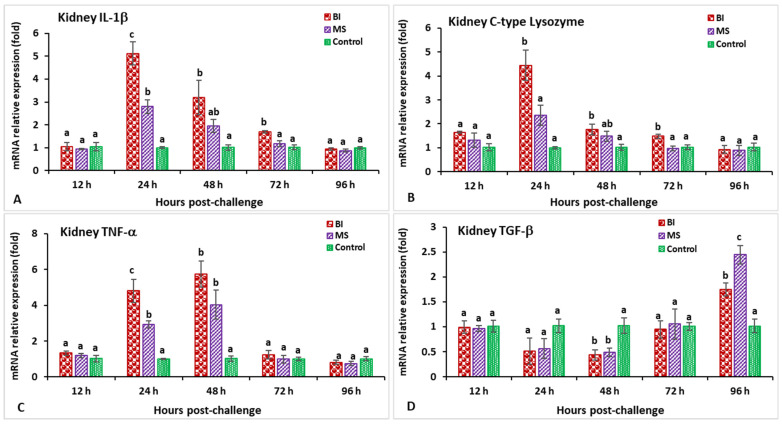
The gene expression of IL-1β (**A**), C-type lysozyme (**B**), TNF-α (**C**), and TGF-β (**D**) in the head kidneys of orally vaccinated fish after being challenged by *Streptococcus iniae* (mean ± SE, *n* = 3). Dissimilar letters denote significant differences (*p* < 0.05).

**Figure 6 biology-11-01268-f006:**
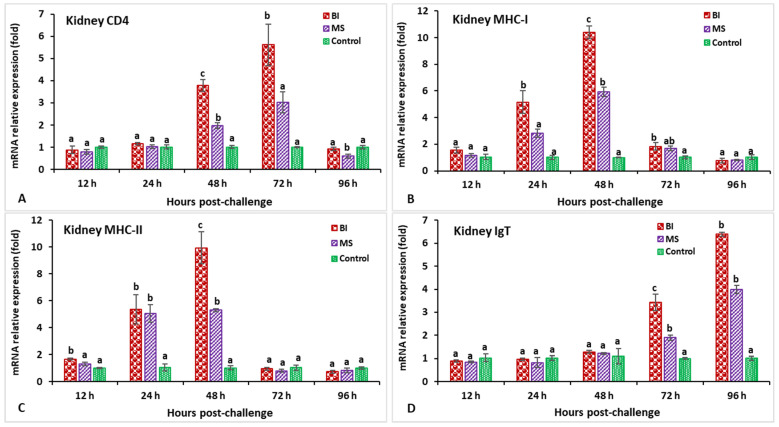
The gene expression of CD4 (**A**), MHC-I (**B**), MHC-II (**C**), and IgT (**D**) in the head kidneys of orally vaccinated fish after being challenged by *Streptococcus iniae* (mean ± SE, *n* = 3). Dissimilar letters denote significant differences (*p* < 0.05).

**Figure 7 biology-11-01268-f007:**
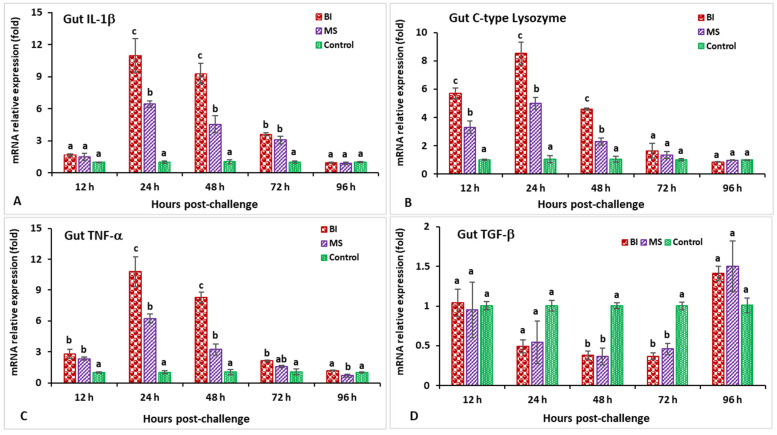
The gene expression of IL-1β (**A**), C-type lysozyme (**B**), TNF-α (**C**), and TGF-β (**D**) in the hindgut of orally vaccinated fish after being challenged by *Streptococcus iniae* (mean ± SE, *n* = 3). Dissimilar letters denote significant differences (*p* < 0.05).

**Figure 8 biology-11-01268-f008:**
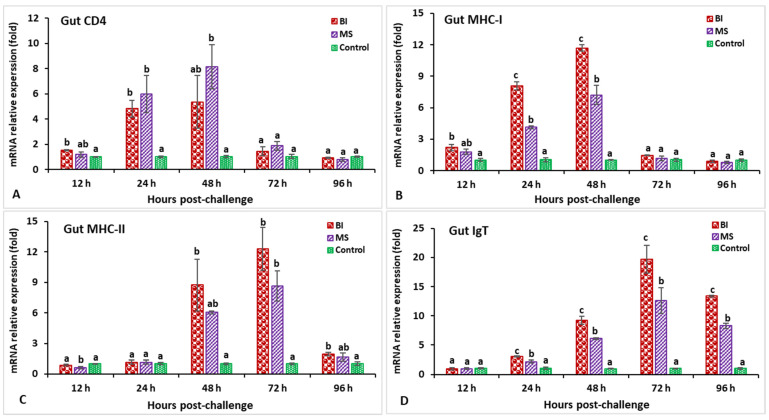
The gene expression of CD4 (**A**), MHC-I (**B**), MHC-II (**C**), and IgT (**D**) in the hindgut of orally vaccinated fish after being challenged by *Streptococcus iniae* (mean ± SE, *n* = 3). Dissimilar letters denote significant differences (*p* < 0.05).

**Figure 9 biology-11-01268-f009:**
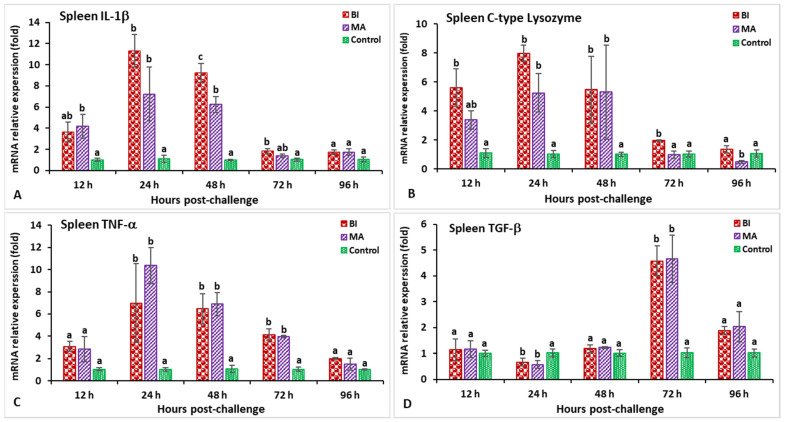
The gene expression of IL-1β (**A**), C-type lysozyme (**B**), TNF-α (**C**), and TGF-β (**D**) in the spleens of orally vaccinated fish after being challenged by *Aeromonas hydrophila* (mean ± SE, *n* = 3). Dissimilar letters denote significant differences (*p* < 0.05).

**Figure 10 biology-11-01268-f010:**
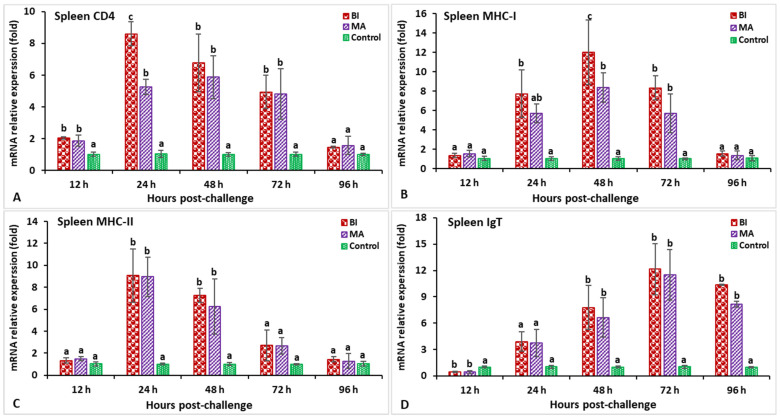
The gene expression of CD4 (**A**), MHC-I (**B**), MHC-II (**C**), and IgT (**D**) in the spleens of orally vaccinated fish after being challenged by *Aeromonas hydrophila* (mean ± SE, *n* = 3). Dissimilar letters denote significant differences (*p* < 0.05).

**Figure 11 biology-11-01268-f011:**
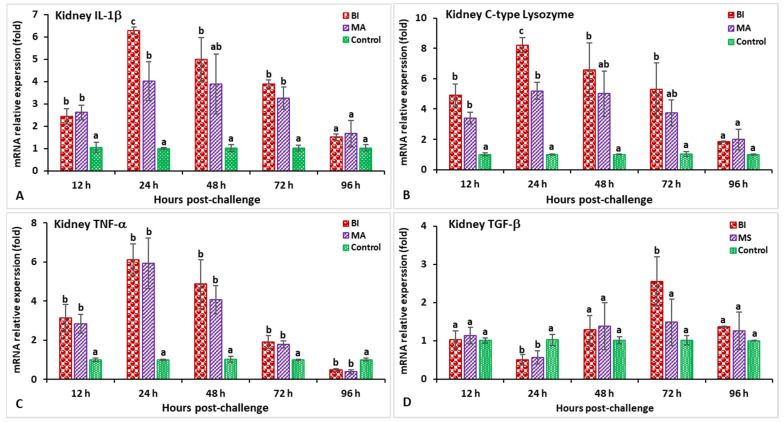
The gene expression of IL-1β (**A**), C-type lysozyme (**B**), TNF-α (**C**), and TGF-β (**D**) in the head kidneys of orally vaccinated fish after being challenged by *Aeromonas hydrophila*. Samples were analyzed in triplicate (3 pools, 2 fish head kidneys/pool) and shown as the mean ± SE. Dissimilar letters denote significant differences (*p* < 0.05).

**Figure 12 biology-11-01268-f012:**
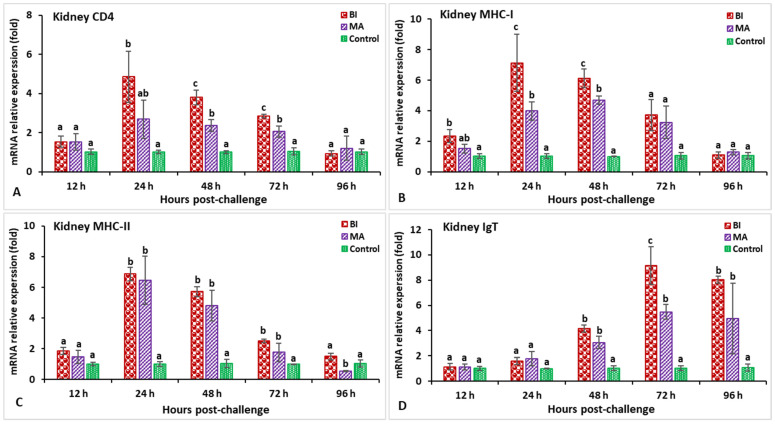
The gene expression of CD4 (**A**), MHC-I (**B**), MHC-II (**C**), and IgT (**D**) in the head kidneys of orally vaccinated fish after being challenged by *Aeromonas hydrophila* (mean ± SE, *n* = 3). Dissimilar letters denote significant differences (*p* < 0.05).

**Figure 13 biology-11-01268-f013:**
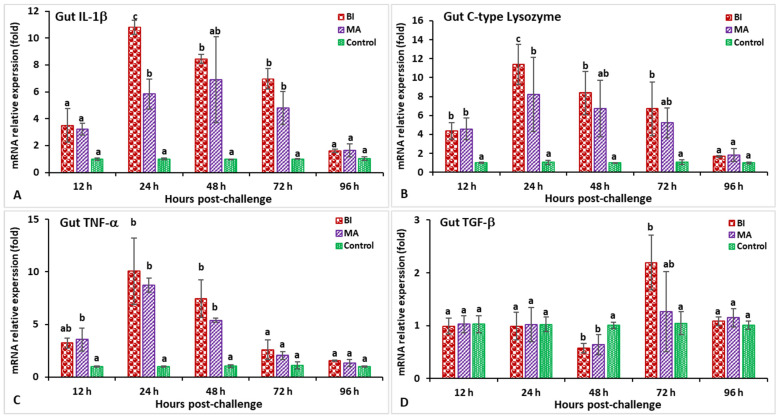
The gene expression of IL-1β (**A**), C-type lysozyme (**B**), TNF-α (**C**), and TGF-β (**D**) in the hindgut tissue of orally vaccinated fish after being challenged by *Aeromonas hydrophila* (mean ± SE, *n* = 3). Dissimilar letters denote significant differences (*p* < 0.05).

**Figure 14 biology-11-01268-f014:**
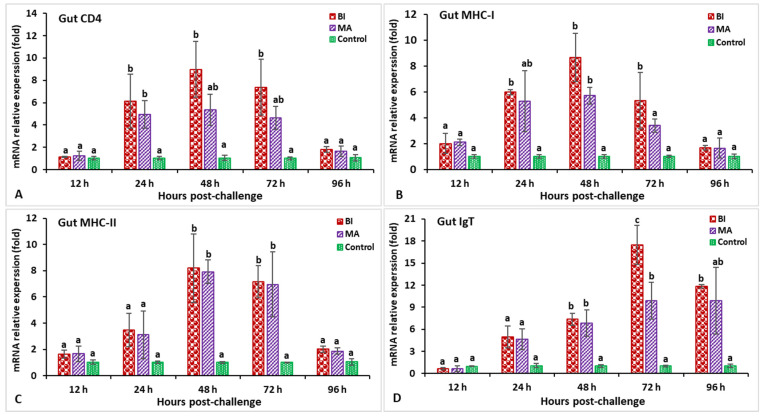
The gene expression of CD4 (**A**), MHC-I (**B**), MHC-II (**C**), and IgT (**D**) in the hindgut tissue of orally vaccinated fish after being challenged by *Aeromonas hydrophila* (mean ± SE, *n* = 3). Dissimilar letters denote significant differences (*p* < 0.05).

**Figure 15 biology-11-01268-f015:**
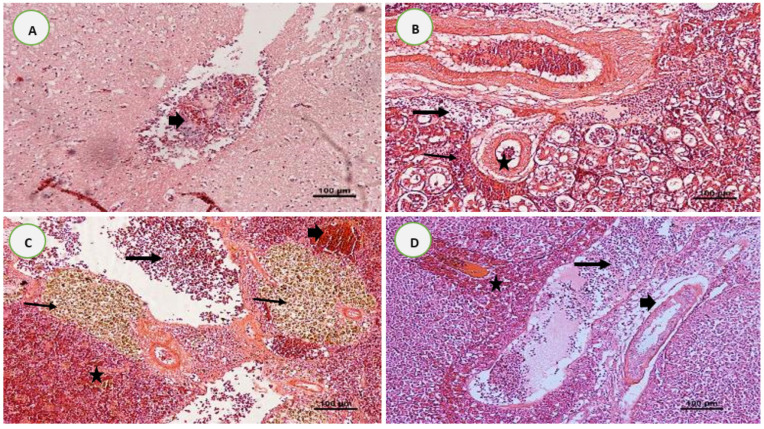
Histological findings in the organs of red tilapia post-challenge with *Streptococcus iniae* (**A**,**B**) and *Aeromonas hydrophila* (**C**,**D**), respectively. (**A**) Brain tissues of unvaccinated fish showing severely congested blood vessels surrounded by mononuclear inflammatory cells and empty space (arrowhead); (**B**) tubular degeneration (star), mononuclear cell infiltration (thin arrow), and a necrotic area between the renal tubules (thick arrow) in the kidney tissues of unvaccinated fish; (**C**) spleen tissue showing severe necrosis and depletion of lymphocytes (thick arrow), presence of hemosiderosis (arrowhead), hemorrhage (star), and huge accumulation of melanomacrophage centers (thin arrow) in unvaccinated fish; and (**D**) liver tissue from the unvaccinated fish showing marked necrosis of hepatocytes (thick arrow) and pancreatic cells (star), along with diffuse vacuolation (arrowhead). H & E, 100X.

**Table 1 biology-11-01268-t001:** Primers for gene expression in red hybrid tilapia by real-time PCR, with expected amplicon size and original sources.

Target mRNA	Sequence (5′-3′)	Amplicon Size (bp)	Accession	References
IL-1B-F	CAAGGATGACGACAAGCCAACC	149	XM_019365844.2	Qiang et al. [26]
IL-1B-R	AGCGGACAGACATGAGAGTGC			
C-type lysozyme-F	AAGGGAAGCAGCAGCAGTTGTG	151	XM_019361339.1	Qiang et al. [26]
C-type lysozyme-R	CGTCCATGCCGTTAGCCTTGAG			
TNF-α-F	GGAAGCAGCTCCACTCTGATGA	137	NM_001279533.1	Qiang et al. [26]
TNF-α-R	CACAGCGTGTCTCCTTCGTTCA			
TGF-β-F	TGCGGCACCCAATCACACAAC	105	XM_025897821.1	Wang et al. [6]
TGF-β-R	GTTAGCATAGTAACCCGTTGGC			
MHC-I-F	TTCTCACCAACAATGACGGG	188	XM_019355579.2	Zhang et al. [27]
MHC-I-R	AGGGATGATCAGGGAGAAGG			
MHC-II-F	AGTGTGGGGAAGTTTGTTGGAT	207	NM_001279562.1	Yao et al. [28]
MHC-II-R	ATGGTGACTGGAGAGAGGCG			
CD4-F	TTCAGTGGCACTTTGCTCCTAA	277	XM_005455490.4	Yao et al. [28]
CD4-R	TGGGCGATGATTTCCAACA			
IgT-F	TCCCACACACTGACCTGTAC	151	XM_025904470.1	Velázquez et al. [29]
IgT-R	GGCCTTGGACTGACTGAGAA			
Control gene		111	NM_001101.5	Qiang et al. [26]
*β*-Actin-F	CCACACAGTGCCCATCTACGA			
*β*-Actin-R	CCACGCTCTGTCAGGATCTTCA			

## Data Availability

Not applicable.

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
