# Peer review of "Effect of an Oral Bivalent Vaccine on Immune Response and Immune Gene Profiling in Vaccinated Red Tilapia (*Oreochromis* spp.) during Infections with *Streptococcus iniae* and *Aeromonas hydrophila"

_biology, 2022, doi:10.3390/biology11091268_

Round 1

Reviewer 1 Report

This MS is a really good piece of research with a substantial amount of work, and well presented and of relevance to bot the academia and the fish farming and vaccine production industries.

The main recommendation to the authors is that I could not find an specific figure so I'm not sure if the vaccines were effective protecting the fish or not and how potent was this vaccine. Can the authors please confirm this? If the vaccine were able to confer or not protection, this should be disclosed in the results by means of presenting survival curves with statistics  and RPS values at the end of the experiment.

If the vaccine(s) was/were effective protecting this should be highlighted as the main result and then all the gene expressions results and final molecule expression analysis (ELISAS for immunoglobulines ) presented as a back up of such (correlates of protection?)

Other minor corrections required are:

Abstract: please introduce the meaning of your acronym BI and BS since the first time these are mentioned, you do fir BI in line 23 so why not doing it since the first time both are mentioned? Line 48 change were for had? 

Introduction.  In line 70 remove "however". Lines 83-91 you refer only negatively about the use of oil adjuvants however its not always the case that these affect tilapia please refer to  https://doi.org/10.1111/jfd.13041 as positive example of immune activation without side effects in tilapia. Also the Intro is already quite lengthy so it requires re focusing please move this section to the discussion and also lines 104-110.#

M&M In line 121, is the term serotype II correct? you never refer to it as such in the cited reference Rahmatullah et al. 

If the authors could please correct the above I would be happy to endorse publication of this MS in this journal  

Regards

Author Response

Cover letter

On behalf of all the authors I, Md Shirajum Monir states that there is no conflict of interest about the study submitted to the journal for possible publication.

Many thanks

Sequel to your mail to us on the review of our manuscript title “Effect of oral bivalent vaccine on immune response and immune gene profiling in vaccinated red tilapia (Oreochromis spp.) during infections with Streptococcus iniae and Aeromonas hydrophila” the following correction and rebuttals were made:

Comments and suggestions from the reviewer (Reviewer 1-First revision)

General comments

I could not find an specific figure so I'm not sure if the vaccines were effective protecting the fish or not and how potent was this vaccine. Can the authors please confirm this? If the vaccine were able to confer or not protection, this should be disclosed in the results by means of presenting survival curves with statistics and RPS values at the end of the experiment.

If the vaccine(s) was/were effective protecting this should be highlighted as the main result and then all the gene expressions results and final molecule expression analysis (ELISAS for immunoglobulines) presented as a back up of such (correlates of protection?)

  • Thank you for your comments. This current study is a sequel to our previous study (Monir et al., 2021). In our previous study, the bivalent vaccine was found to give a high protection to the host with a relative percent survival (RPS) of 82.22 ± 3.85% when challenged with iniae, 77.78 ± 3.85% when challenged with A. hydrophila and 77.78 ± 3.85% when co- challenged with both S. iniae and A. hydrophila, which were significantly higher (P < 0.05) compared to the other groups. The bivalent vaccinated group also showed significantly (P < 0.05) higher partial cross-protections following challenges with S. agalactiae (RPS at 60.00 ± 6.67%) and A. veronii (RPS at 57.78 ± 7.70%). The protective efficacies of the bivalent vaccine is mentioned in Line 109

Abstract

Please introduce the meaning of your acronym BI and BS since the first time these are mentioned, you do for BI in line 23 so why not doing it since the first time both are mentioned? Line 48 change were for had? 

  • Thank you for your comments. The meaning of the acronyms were mentioned in the abstract and in Line 48 the word “were” was changed to “had”.

Introduction. 

In line 70 remove "however". Lines 83-91 you refer only negatively about the use of oil adjuvants however its not always the case that these affect tilapia please refer to  https://doi.org/10.1111/jfd.13041 as positive example of immune activation without side effects in tilapia. Also the Intro is already quite lengthy so it requires re focusing please move this section to the discussion and also lines 104-110.#

  • Thank you for your comments. The word “however” was removed.
  • Lines 83-91 was revised
  • The introduction was shorten following reviewer’s recommendation
  • Line 104-110 was revised.

Materials and Methods

In line 121, is the term serotype II correct? you never refer to it as such in the cited reference Rahmatullah et al. 

  • Thank you for your comments. The term “serotype II” was removed to avoid confusion.

 Reference

Monir, M.S., Yusoff, M.S.M., Zulperi, Z.M., Hassim, H.A., Zamri-Saad, M., Amal, M.N.A., Salleh, A., Mohamad, A., Yie, L.J., Ina-Salwany, M.Y., 2021. Immuno-protective efficiency of feed-based whole-cell inactivated bivalent vaccine against Streptococcus and Aeromonas infections in red hybrid tilapia (Oreochromis niloticus × Oreochromis mossambicus). Fish Shellfish Immunol. 113, 162–175. https://doi.org/10.1016/j.fsi.2021.04.006

Reviewer 2 Report

The paper described  the IgM response to the oral bivalent vaccine and profiling of IL-1β, TNF-α, C-type lysozyme, TGF-β, CD4, MHC-I, MHC-II and IgT genes in red tilapia following challenged with  Streptococcus iniae and Aeromonas hydrophila. The results showed that compared to the monovalent and control groups, the expression levels of different innate and adaptive immune genes were significantly higher in both systemic and mucosal immune organs of the bivalent vaccinated fish. The oral bivalent incorporated (BI) vaccine could induce significant humoral and cellular immunity and provide strong protections against virulent S. iniae and A. hydrophila infections. The experiment is well designed and the result is interesting that will provoke the development of suitable vaccine against  virulent S. iniae and A. hydrophila later. However, there are some expression errors and linguistic mistakes in the manuscript, which should corrected and refined by native English experts before it is accepted for publication.  

Author Response

Cover letter

On behalf of all the authors I, Md Shirajum Monir states that there is no conflict of interest about the study submitted to the journal for possible publication.

Many thanks

Sequel to your mail to us on the review of our manuscript title “Effect of oral bivalent vaccine on immune response and immune gene profiling in vaccinated red tilapia (Oreochromis spp.) during infections with Streptococcus iniae and Aeromonas hydrophila” the following correction and rebuttals were made:

Comments and suggestions from the reviewer (Reviewer 1-First revision)

General comments

  1. There are some expression errors and linguistic mistakes in the manuscript, which should be corrected and refined by native English experts before it is accepted for publication.  

  • The errors and linguistic mistakes in the manuscript have been revised, and all specified changes were accepted. We apologize for the numerous errors in punctuation and tenses. This manuscript was later revised by a senior lecturer with better English writing skills. We appreciate the patience of the Editor in meticulously correcting these.
